# Pravastatin and Gemfibrozil Modulate Differently Hepatic and Colonic Mitochondrial Respiration in Tissue Homogenates from Healthy Rats

**DOI:** 10.3390/cells8090983

**Published:** 2019-08-27

**Authors:** Anna Herminghaus, Eric Laser, Jan Schulz, Richard Truse, Christian Vollmer, Inge Bauer, Olaf Picker

**Affiliations:** Department of Anesthesiology, University Hospital Duesseldorf, Moorenstrasse 5, 40225 Duesseldorf, Germany

**Keywords:** pravastatin, gemfibrozil, liver, colon, mitochondrial function

## Abstract

Statins and fibrates are widely used for the management of hypertriglyceridemia but they also have limitations, mostly due to pharmacokinetic interactions or side effects. It is conceivable that some adverse events like liver dysfunction or gastrointestinal discomfort are caused by mitochondrial dysfunction. Data about the effects of statins and fibrates on mitochondrial function in different organs are inconsistent and partially contradictory. The aim of this study was to investigate the effect of pravastatin (statin) and gemfibrozil (fibrate) on hepatic and colonic mitochondrial respiration in tissue homogenates. Mitochondrial oxygen consumption was determined in colon and liver homogenates from 48 healthy rats after incubation with pravastatin or gemfibrozil (100, 300, 1000 μM). State 2 (substrate dependent respiration) and state 3 (adenosine diphosphate: ADP-dependent respiration) were assessed. RCI (respiratory control index)—an indicator for coupling between electron transport chain system (ETS) and oxidative phosphorylation (OXPHOS) and ADP/O ratio—a parameter for the efficacy of OXPHOS, was calculated. Data were presented as a percentage of control (Kruskal–Wallis + Dunn’s correction). In the liver both drugs reduced state 3 and RCI, gemfibrozil-reduced ADP/O (complex I). In the colon both drugs reduced state 3 but enhanced ADP/O. Pravastatin at high concentration (1000 µM) decreased RCI (complex II). Pravastatin and gemfibrozil decrease hepatic but increase colonic mitochondrial respiration in tissue homogenates from healthy rats.

## 1. Introduction

Statins are among the most widely prescribed drug classes in the world. They are used to lower low density lipoprotein-cholesterol (LDL-C) serum levels in patients for the prevention and treatment of cardiovascular diseases [1]. They exhibit a wide range of effects: additionally to the inhibition of cholesterol synthesis, they modulate inflammatory response, affect coagulation system, induce apoptosis and decrease oxidative stress [2].

If statins are not successful, guidelines recommend peroxisome proliferator-activated receptor alpha (PPARα) agonists (fibrates) for the management of hypertriglyceridemia [3]. Studies have shown that fibrates also have pleiotropic effects like improving endothelial dysfunction [4,5], inhibiting the expression of adhesion molecules and inflammatory cytokines [6] and decreasing oxidative stress and nitric oxide production [5]. Moreover, fibrates can inhibit coagulation [4,7] and improve haemorheologic parameters [8].

However, these agents also have limitations, most importantly due to pharmacokinetic interactions, such as an increased risk of myopathy through a combination of statins and gemfibrozil [9], or side effects, which include digestive disorders, reversible elevation in serum creatinine and liver enzymes [10].

It is conceivable that some of the adverse events like liver dysfunction or gastrointestinal discomfort are caused by mitochondrial dysfunction [11,12]. However, data about effects of statins and fibrates on mitochondrial function in different organs are inconsistent and partially contradictory. Data about effects of these drugs on colonic mitochondria are lacking completely.

Statins can affect skeletal muscle mitochondria in vitro by inhibiting respiratory chain complexes and oxidative capacity [12,13], decreasing mitochondrial membrane potential [13], uncoupling oxidative phosphorylation, inducing mitochondrial swelling and apoptosis [13] and decreasing mitochondrial density [14]. Statins also uncouple state 2 respiration and can inhibit the activity of the complexes of the respiratory chain in liver mitochondria, but the effect is drug-dependent (simvastatin has a strong deteriorating effect, while pravastatin does not seem to affect hepatic mitochondria) [15].

Hydrophilic statins (e.g., pravastatin) are considered as less ‘mitotoxic’ compared to lipophilic statins such as cerivastatin, fluvastatin, atorvastatin and simvastatin [13]. Even if some authors failed to show any effects of pravastatin on mitochondrial respiration in muscle [13], in the liver [16] and in HL-1 cardiomyocytes [1], cases of drug induced hepatotoxicity are reported [17]. Furthermore, evidence exists for even positive effects of statins on mitochondrial function. Bouitbir et al. showed that statins promote mitochondrial function and mitochondrial biogenesis in human heart muscle [18].

Effects of fibrates on mitochondrial function are also not clearly understood. From one side, in vitro results suggest impaired mitochondrial function, via direct inhibition of mitochondrial respiration (mainly complex I) [19], by membrane depolarization [20] and through increases in uncoupled respiration [15,21]. From the other side, fibrates as PPAR-α agonists can enhance mitogenesis and therefore mitochondrial activity [22]. There are also differences among single fibrates concerning effects on mitochondrial function and gemfibrozil seems to be less mitotoxic than the other drugs from this group [19].

Taken together, the effects of statins and fibrates on mitochondrial function in different organs have been insufficiently examined. Data concerning hepatic mitochondria are inconsistent and about other organs like colon are lacking completely. The aim of this study was therefore to investigate the concentration dependent effect of pravastatin (statin) and gemfibrozil (fibrate) on hepatic and colonic mitochondrial respiration in tissue homogenates from healthy rats.

## 2. Materials and Methods

### 2.1. Animals:

The study was approved from the Animal Ethics Committee of the University of Duesseldorf, Germany (project identification code: O27/12), and performed in accordance with the Guide for the Care and Use of Laboratory Animals of the National Institutes of Health. The authors ensured that their research complied with the commonly accepted ‘3Rs’: replacement, reduction and refinement.

Male Wistar rats were purchased from the breeding facilities of the University of Düsseldorf (Düsseldorf Germany) or from Janvier Labs (Le Genest-Saint-Isle, France). They were kept at an artificial 12-h light/dark cycle at constant room temperature and humidity with free access to standard chow and tap water.

Forty eight rats (approximately 3 months old) were sacrificed by decapitation under deep sedation with sodium pentobarbital (90 mg/kg) and liver and colon were harvested.

### 2.2. Preparation of Liver and Colon Homogenates

Liver and colon homogenates were prepared as described previously [23,24,25]. Briefly, liver tissue was placed in 4 °C cold isolation buffer, minced into 2–3-mm^3^ pieces, rinsed twice in isolation buffer to remove traces of blood, and homogenized (Potter-Elvehjem, Pro Scientific, Swedesboro, NJ, USA, 5 strokes, 2000 rpm). Freshly harvested colon tissue was placed in isolation buffer enriched with 2% bovine serum albumin (BSA, Sigma-Aldrich Corporation, St. Louis, MO, USA), longitudinally opened, and dried with a cotton pad to remove remains of faeces and mucus. After incubation with 0.05% trypsin (Thermo Fisher Scientific, Dreieich, Germany) for 5 min, the tissue was transferred into the isolation buffer containing 2% BSA and protease inhibitors (cOmplete™ Protease Inhibitor Cocktail, Roche Life Science, Mannheim, Germany), minced, and finally homogenized (see above). Protein concentration in the tissue homogenates was determined by the Lowry method [26] with bovine serum albumin as a standard. All procedures were performed on ice, all buffer were kept by 4 °C.

### 2.3. Measurement of Mitochondrial Respiratory Rates

Mitochondrial oxygen consumption was measured as described previously [23,24,25]. Briefly, the measurement was performed at 30 °C using a Clark-type electrode (model 782, Strathkelvin instruments, Glasgow, Scotland). Tissue homogenates were suspended in respiration medium to yield a protein concentration of 4 mg/mL or 6 mg/mL for liver and colon, respectively.

#### 2.3.1. Mitochondrial State 2 Respiration

Mitochondrial state 2 respiration was performed in the presence of either complex I substrates glutamate (Fluka, München, Germany) and malate (Serva Electrophoresis GmbH, Heidelberg, Germany) (both 2.5 mM, G–M) or the complex II substrate succinate (Sigma-Aldrich Corporation, St. Louis, MO, USA) (10 mM for liver, 5 mM for colon, S) combined with 0.5 µM rotenone (Sigma-Aldrich Corporation, St. Louis, MO, USA)—the inhibitor of complex I activity. Rotenon was always added before addition of succinate.

#### 2.3.2. Mitochondrial State 3 Respiration

The maximal coupled mitochondrial respiration (state 3) was measured after the addition of adenosine diphosphate—ADP (Sigma-Aldrich Corporation, St. Louis, MO, USA) (250 μM for liver, 50 μM for colon). The respiratory control index (RCI) was calculated (state 3/state 2) to define the coupling between the electron transport system (ETS) and oxidative phosphorylation (OXPHOS). To reflect the efficacy of OXPHOS, the ADP/O ratio was calculated from the amount of ADP added and oxygen consumed in state 3. The average oxygen consumption was calculated as the mean from three technical replicates.

#### 2.3.3. General Conditions

The solubility of oxygen was assumed to be 223 μmol O_2_/L at 30 °C according to the Strathkelvin instruments manual. Respiration rates were expressed as nmol/min/mg protein. No correction of the natural drift of the electrode was made since a drift of less than 0.5% over 12 h is neglectable in our experimental setup.

#### 2.3.4. Quality Control for the Preparation Procedure

At the beginning of every experiment mitochondria were checked for leakage by the addition of 2.5 μM cytochrome c (Sigma-Aldrich Corporation, St. Louis, MO, USA) and 0.05 μg/mL oligomycin (Calbiochem by Merck KGaA, Darmstadt, Germany) at the beginning of every experiment. There was no increase in flux after the addition of cytochrome c, indicating integrity of the mitochondrial outer membrane. When ATP synthesis was inhibited by oligomycin, the mitochondria were transferred to state 2, which reflects the respiration rate compensating the proton leak. The lack of difference in O_2_ consumption after adding oligomycin compared to state 2 indicates that the inner membrane was intact and mitochondria were not damaged through the preparation procedure.

#### 2.3.5. Experimental Conditions

The assessment of mitochondrial respiration was performed after addition of carrier substance (distilled water)—control, or different concentrations of pravastatin (100 µM, 300 µM and 1000 µM). For experiments with gemfibrozil (100 µM, 300 µM and 1000 µM), dimethyl sulfoxide (DMSO) (Sigma-Aldrich Corporation, St. Louis, MO, USA) 0.5%, 1.5% and 5% respectively were used as controls. The incubation took place at room temperature (kept at 21 °C) for 3 min. Eight biological and three technical (three separate measurements from a single homogenate) replicates were performed.

### 2.4. Statistical Analysis

Statistical analysis was performed with GraphPad Prism 8.0 (GraphPad Software, GraphPad Software, San Diego, CA, USA). After checking the data set for normality (Kolmogorov–Smirnov) a Kruskal–Wallis test of variance followed by a post hoc Dunn’s correction were performed. Means ± standard deviations (S.D.) were determined. Data are presented as percentage of control values, *P* < 0.05 was considered significant. 

## 3. Results

### 3.1. Effects of Pravastatin on Hepatic Mitochondrial Respiration

The lowest concentration of pravastatin (100 µM) did not affect the mitochondrial respiration in the liver. Pravastatin in higher concentrations (300 µM and 1000 µM) reduced state 3 (complex I: pravastatin 300 µM: 77.5% ± 3.2%*, pravastatin 1000 µM: 60.7% ± 5.3%*; complex II: pravastatin 300 µM: 84.8% ± 2.4%*, pravastatin 1000 µM: 72.3% ± 6.2%*) (Figure 1A,D) and RCI (complex I: pravastatin 300 µM: 75.3% ± 5.0%*, pravastatin 1000 µM: 66.4% ± 4.2%*, complex II: pravastatin 300 µM: 82.6% ± 6.1%*, pravastatin 1000 µM: 72.2% ± 5.9%*) (Figure 1B,E) for both complexes without changing the ADP/O-ratio (Figure 1C,F).

### 3.2. Effects of Pravastatin on Colonic Mitochondrial Respiration

The lowest concentration of pravastatin (100 µM) did not affect the mitochondrial respiration in the colon. Pravastatin in middle concentration (300 µM) reduced state 3 for complex II (86.1% ± 4.8%*) (Figure 2D) and in high concentration (1000 µm) decreased state 3 for both complexes (complex I: 63.6% ± 8.2%*, complex II: 77.3% ± 4.1%*) (Figure 2A,D).

The RCI in colonic mitochondria was decreased only by pravastatin at the highest concentration (1000 µM) and only for complex II (83.8 ± 12.6%*) (Figure 2E).

Pravastatin in high concentration (1000 µM) increased the ADP/O-ratio for both complexes (complex I: 151.5% ± 50.4%*, complex II: 136.4% ± 24.9%*) (Figure 2C,F).

### 3.3. Effects of Gemfibrozil on Hepatic Mitochondrial Respiration

Similarly to pravastatin, gemfibrozil in low concentration (100 µM) did not affect the mitochondrial respiration in the liver. Gemfibrozil at higher concentrations (300 µM and 1000 µM) reduced state 3 (complex I: gemfibrozil 300 µM: 56.9% ± 5.5%*, gemfibrozil 1000 µM: 24.4% ± 2.5%*; complex II: gemfibrozil 300 µM: 87.8% ± 18.5%*, gemfibrozil 1000 µM: 50.9% ± 11.4%*) (Figure 3A,D) and RCI (complex I: gemfibrozil 300 µM: 45.1% ± 5.2%*, gemfibrozil 1000 µM: 21.5% ± 2.8%*, complex II: gemfibrozil 300 µM: 70.4% ± 5.5%*, gemfibrozil 1000 µM: 37.8% ± 8.3%*) (Figure 3B,E) for both complexes. Gemfibrozil 300 µM reduced ADP/O for complex I (81.5% ± 7.6%*) (Figure 3C). ADP/O-ratio after treatment with gemfibrozil 1000 µM was significantly higher than ADP/O-ratio in other gemfibrozil-groups.

### 3.4. Effects of Gemfibrozil on Colonic Mitochondrial Respiration

Gemfibrozil in low concentration (100 µM) reduced state 3 for complex I (89.3% ± 5.5%*). Gemfibrozil at the middle concentration (300 µM) did not show any effect on colonic mitochondrial respiration. Gemfibrozil at high concentration (1000 µM) reduced state 3 for both complexes (complex I: 80.8% ± 6.6*, complex II: 92.5% ± 15.2%*) and increased the ADP/O-ratio for complex I (133.8% ± 33.8%*) without changing the RCI (Figure 4B,E).

## 4. Discussion

The main result of this study is that the effect of pravastatin and gemfibrozil is organ specific and dose dependent. Both drugs seem to have a deteriorating effect on hepatic mitochondria but rather positive influence on colonic mitochondrial respiration.

The chosen experimental setting is based on our previous study [24]. The drug concentrations correspond to the literature describing similar in vitro experiments [13,15,19,27]. The measurements are performed at 30 °C which is a methodological standard [13,15,20], but not a physiological condition. Thus, the data may not reflect full effects of the drugs in vivo. While liver is mainly composed of hepatocytes, colon consists of different cell lines like epithelial cells, smooth muscle cells, adipocytes and many others. Therefore, our results cannot relate to a special cell line.

In hepatic mitochondria, pravastatin dose dependently reduced ADP-induced mitochondrial respiration-state 3, and coupling between electron transport chain (ETS) and oxidative phosphorylation (OXPHOS)-RCI, without changing the efficacy of oxidative phosphorylation-ADP/O.

In colonic mitochondria, pravastatin also reduced state 3 and RCI, however, to a minor extent, mainly at the higher concentration and preferably through complex II. In contrast to hepatic mitochondria, pravastatin increased the efficacy of OXPHOS in the colon for both complexes.

Our results concerning pravastatin and mitochondrial respiration are new findings compared to the results of other authors who mainly could not show any effects of this drug on mitochondrial respiration. Marques et al. examined the effect of pravastatin on hepatic mitochondrial in LDL knockout mice after oral pretreatment with 40 mg/kg pravastatin and did not observe any changes either [16]. Sugiyama et al. tested the effects of pravastatin on age-related changes in mitochondrial function in rats after long-term therapy. They could show that pravastatin significantly accelerated the age-related decline in the activity of complex I of diaphragm mitochondria, whereas the aging effect on mitochondrial respiratory function was not observed on heart muscle and liver. Pravastatin did not significantly affect cardiac and hepatic respiratory function [28]. Kaufmann et al. [13] did not observe any effects of pravastatin on mitochondrial oxygen consumption in isolated muscle mitochondria in vitro using similar pravastatin concentrations (50–400 µM). Godoy et al. [1] tested the influence of atorvastatin and pravastatin (10 µM) on HL-1 cardiomyocyte mitochondrial function and could show, that atorvastatin altered mitochondrial function compared to cardiomyocytes treated with pravastatin. The difference between our results (declined mitochondrial respiration in liver) and the other findings (unchanged mitochondrial respiration) could be caused by many factors like different experimental conditions (in vitro vs. ex vivo, different drug concentrations in in vitro experiments and oral pretreatment), long-term therapy vs. single dose and different tissues (liver, muscle, cardiomyocytes). It is well known, that mitochondrial function varies between organs [29].

The effect of pravastatin on colonic mitochondria was different from that in the liver. The drug moderately reduced state 3 and RCI, mainly in high concentration and preferably through complex II but increased the efficacy of OXPHOS for both complexes. It seems to be a rather positive effect reflected in a higher efficacy of OXPHOS with reduced mitochondrial respiration. In general, an increase in mitochondrial respiration is considered as an improvement and vice versa. However, mitochondria may also dynamically respond to specific conditions, and changes in oxphos capacity or RCI may simply reflect a response rather than an improvement or impairment. Under physiological conditions with sufficient oxygen supply, the impact on cell metabolism is probably of minor relevance. However, this effect might gain a major importance as an adaptive response under compromised conditions associated with e.g., cellular hypoxia. To clarify, whether this observation is favorable for the cell metabolism, the underlying processes like activity of the single complexes of the respiratory chain, mitochondrial membrane potential or tissue ATP-concentration need to be further analyzed.

To our best knowledge, there are no data about influence of pravastatin on mitochondrial function in the colon so we cannot refer our results to those of other authors.

For pravastatin, plasma concentrations after oral administration of 20–40 mg/day are in the range of 0.1–0.229 µM [30] which is considerably lower than those concentrations applied in our in vitro study. The oral bioavailability of pravastatin is low (17%) because of incomplete absorption and a first-pass effect. The drug is rapidly absorbed from the upper part of the small intestine, probably via proton-coupled carrier-mediated transport, and then taken up by the liver by a sodium-independent bile acid transporter [31,32]. The uptake of the drugs into different tissues differs substantially and leads to wide ranges of drug concentrations in the target organs. Yamazaki et al. [33] examined the pharmacokinetic properties of pravastatin after intravenous and portal vein application in rats and could show that the largest clearance was observed for the liver, followed by the kidney whereas other tissues like small intestine exhibited only a minor uptake. Hatanaka et al. showed that pravastatin accumulated in the liver and reached even higher concentrations compared to plasma levels. Interestingly, the pravastatin plasma concentration increased with increasing dose, whereas the contrary was the case in liver and small intestine [34]. We chose relatively high drug concentration according to the similar in vitro experiments described in the literature [13,17]. Nevertheless, our results may be relevant for the in vivo situation since the most adverse events with statins have been described in patients having a drug–drug interaction leading to a higher drug concentration or underlying mitochondrial disease rendering them more sensitive to statins [35,36]. Moreover, many patients suffer from multimorbidity. Renal and/or liver insufficiency can also result in supraclinical or even toxic drug plasma concentrations.

The effects of gemfibrozil on hepatic mitochondrial respiration were similar to pravastatin but seem to be more pronounced. Gemfibrozil reduced dose dependently state 3 and RCI for both complexes and ADP/O-ratio for complex I. However, the increase in ADP/O ratio after treatment with gemfibrozil 1000 µM most likely reflects rather a sign of terminal uncoupling than improvement of efficacy of OXPHOS. To test this hypothesis, we treated hepatic mitochondria with an uncoupler—2-[2-(3-Chlorophenyl)hydrazinylyidene]propanedinitrile (CCCP)—and there was no further enhancement in mitochondrial respiration confirming the terminal uncoupling.

Our results are consistent with those of other authors, who also observed a decrease in mitochondrial respiration and uncoupling effect of gemfibrozil. Nadanaciva et al. [15] showed that gemfibrozil in concentration of 500 nmol/mg mitochondrial proteins (which would correspond to 2000 µM in our experimental setting) lowered state 3 in isolated hepatic rat mitochondria about more than 50%. Zhou et al. [20] examined the effects of different fibrates on mitochondrial bioenergetics and showed that gemfibrozil at 75 µM uncouples isolated hepatic rat mitochondria and reduces the efficacy of the OXPHOS. Zhou et al. suggest that the underlying mechanism of uncoupling could be the induction of mitochondrial permeability transition pores.

In the colon, gemfibrozil decreased slightly ADP-dependent mitochondrial respiration, but did not affect the coupling between ETS and OXPHOS and improved the efficacy of OXPHOS for complex I. Similar to pravastatin, we presume that gemfibrozil could also have a protective effect on colonic mitochondria, allowing an efficient ATP-production by reduced mitochondrial oxygen consumption. Also in this case, this mechanism may be relevant under pathologic conditions like sepsis or hemorrhagic shock, but probably does not play a pivotal role when the oxygen supply is sufficient. Nevertheless, mechanisms compensating a lack of oxygen are substantially important in organs like the colon. When circulation becomes unstable, e.g., in septic shock, blood flow is redistributed to maintain the oxygenation of vital organs as heart or brain, while microcirculation in less essential organs like the splanchnic region, kidney and liver is critically reduced [37]. We are the first to analyze the effects of gemfibrozil on colonic mitochondrial function, so no comparison with other results can be made.

As described above, we used in our experiment higher gemfibrozil concentrations than clinically occur. After the standard oral administration of 1200 mg/day, the plasma concentration of gemfibrozil reaches 10–20 mg/L (40–80 µM) [38,39]. Also in this case we consider our result as clinically relevant, because fibrates are often combined with other lipid-lowering drugs like statins or thiazolidinediones and the combination can lead to higher plasma levels. Moreover, the clinically used drug combinations, like statins plus fibrates, show different effects to the single application. In our experiments, pravastatin and gemfibrozil showed similar negative effects on mitochondrial respiration in liver and positive influence on the colonic mitochondria. So it is conceivable, that additive effects are observed with co-incubation. Nadanaciva et al. showed that gemfibrozil at 62 nmol/mg protein (which would correspond to 248 µM in our experimental setting) did not affect mitochondrial function, but in combination with cerivastatin depressed the mitochondrial respiration significantly [15].

## 5. Conclusions

Taken together, we show new findings about organ-specific and concentration-dependent effects of two clinically important and widely used drugs, pravastatin and gemfibrozil, on hepatic and colonic mitochondrial respiration. Results from this study reveal a rather negative effect of both drugs on hepatic mitochondrial respiration. This could possibly be one of the mechanisms contributing to elevated liver enzymes during the therapy with these drugs. This hypothesis must be considered very carefully because we examined the mitochondrial oxygen consumption, which depicts only one aspect of the complex mitochondrial function within a cell. Furthermore, our experiments are conducted in vitro with animal tissues and these adverse effects are not fully understood and are complex processes including many factors like preexisting organ damage and co-medication.

The positive effects of pravastatin and gemfibrozil on colonic mitochondria could contribute to adaptive cell mechanisms under pathological conditions like sepsis or hemorrhagic shock, where tissue hypoxia may occur, allowing better oxygen utilization. Also in this case, the results must be interpreted very cautiously and further research in this field is needed. Our data extend our knowledge about a possible mode of action of both drugs and offer a new insight into conceivable mechanisms of their side effects.

## Figures and Tables

**Figure 1 cells-08-00983-f001:**
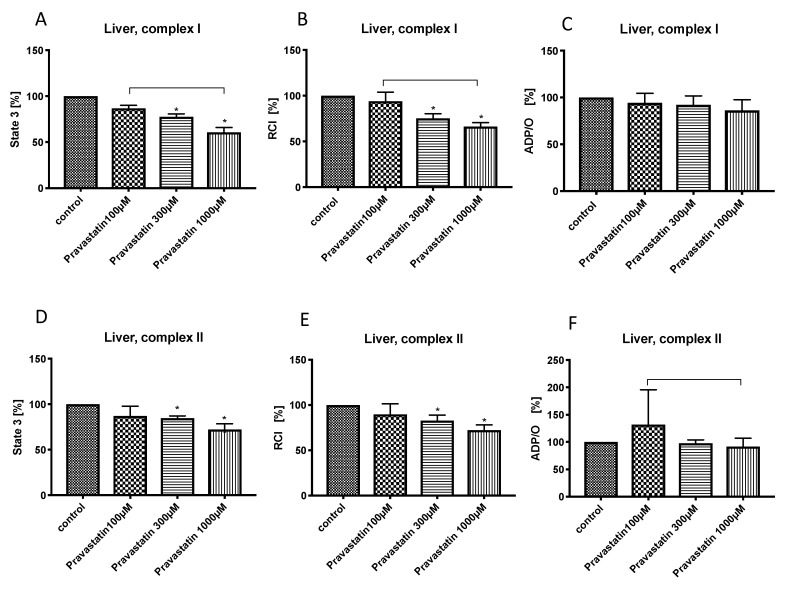
Effect of pravastatin (100 µM, 300 µM and 1000 µM) on hepatic mitochondrial respiration: State 3 for complex I (**A**) and II (**D**), respiratory control index (RCI) for complex I (**B**) und II (**E**) and ADP/O ratio for complex I (**C**) and II (**F**). Data are presented as mean ± standard deviation (S.D.), *n* = 7–8, * *P* < 0.05 vs. control, *P* < 0.05 between groups.

**Figure 2 cells-08-00983-f002:**
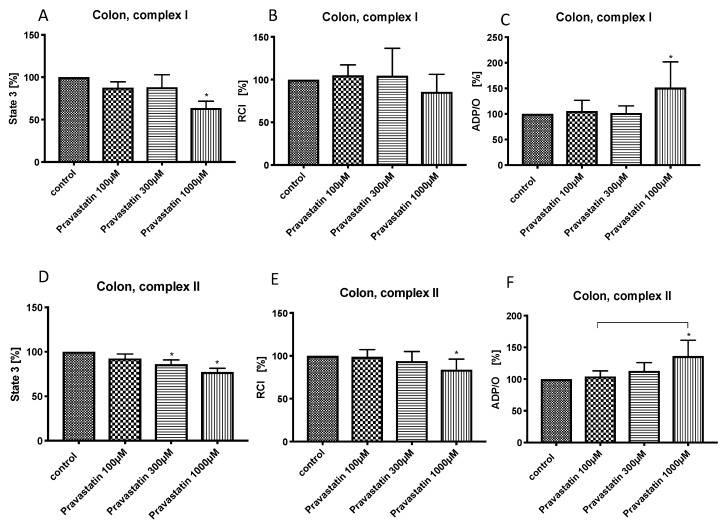
Effect of pravastatin (100 µM, 300 µM and 1000 µM) on colonic mitochondrial respiration: State 3 for complex I (**A**) and II (**D**), respiratory control index (RCI) for complex I (**B**) und II (**E**) and ADP/O ratio for complex I (**C**) and II (**F**). Data are presented as mean ± S.D., *n* = 7–8, * *P* < 0.05 vs. control, *P* < 0.05 between groups.

**Figure 3 cells-08-00983-f003:**
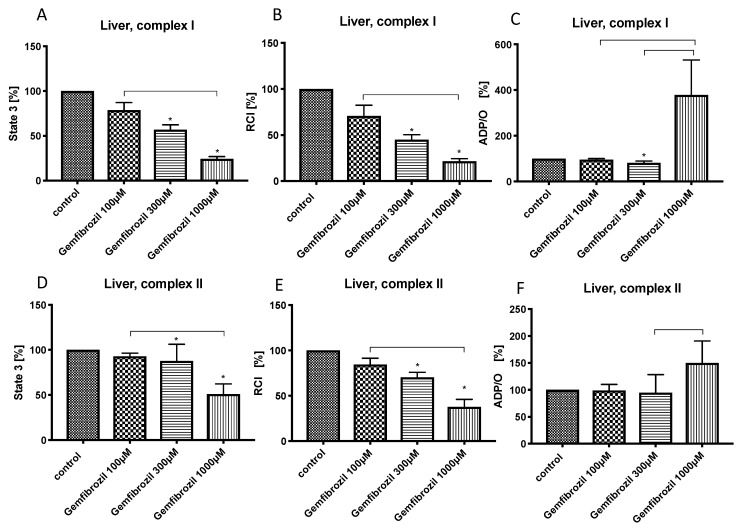
Effect of gemfibrozil (100 µM, 300 µM and 1000 µM) on hepatic mitochondrial respiration: State 3 for complex I (**A**) and II (**D**), respiratory control index (RCI) for complex I (**B**) und II (**E**) and ADP/O ratio for complex I (**C**) and II (**F**). Data are presented as mean ± S.D., *n* = 7–8, * *P* < 0.05 vs. control, *P* < 0.05 between groups.

**Figure 4 cells-08-00983-f004:**
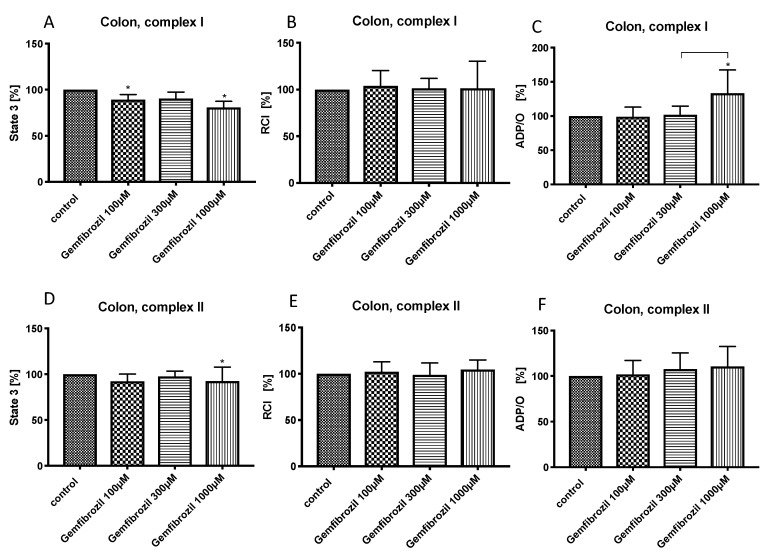
Effect of gemfibrozil (100 µM, 300 µM and 1000 µM) on colonic mitochondrial respiration: State 3 for complex I (**A**) and II (**D**), respiratory control index (RCI) for complex I (**B**) und II (**E**) and ADP/O ratio for complex I (**C**) and II (**F**). Data are presented as mean ± S.D., *n* = 7–8, * *P* < 0.05 vs. control, *P* < 0.05 between groups.

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
