# Peer review of "Pravastatin and Gemfibrozil Modulate Differently Hepatic and Colonic Mitochondrial Respiration in Tissue Homogenates from Healthy Rats"

_cells, 2019, doi:10.3390/cells8090983_

Round 1

Reviewer 1 Report

The article presents data on mitochondrial function under in-vitro exposure to Pravastatin and Gemfibrozil. The study is based on standard methods of respirometry and technically well conducted. In my opinion, several limitations need to be addressed more deeply:

Respirometry has been conducted at 30°C, which obviously does not correspond to physiological conditions. It cannot be simply assumed that the effects of temperature are equal for all parts of the respiratory chain. Therefore, the data may not reflect the effects of the drugs in-vivo. This limitation needs to be addressed and discussed. In contrast to the liver, which is mainly composed by hepatocytes, tissue samples from the colon are probably less homogeneous regarding the cell type. Are data available on the composition of colon tissue homogenates? This should be discussed. It is not clear, why liver and colon were investigated, but not tissues previously studied by other authors, This could allow to better compare the data. In that sense, the introduction should be improved. Similarly, it is not clear why Pravastatin and gemfibrozil only were tested. It should be avoided to speak of improvement or impairment of mitochondrial function. First of all, mitochondria have several functions within the cell. In this study, however, only mitochondrial respiration has been studied. In general, an increase in mitochondrial respiration is considered as an improvement and vice versa. However, mitochondria may also dynamically respond to specific condition, and changes in oxphos capacity or RCR may simply reflect a response rather than an improvement or impairment.

Author Response

Dear Editor, dear Reviewers,

We sincerely thank the Reviewers for evaluating our submission and their constructive criticism. We addressed all comments and have revised our paper accordingly. After implementing the Reviewer’s points we think the manuscript has greatly improved.

Additionally, we revised the manuscript carefully concerning the repetition rate with our previous publications as suggested by the Assistant Editor.

Changes made in the manuscript are highlighted in yellow in the revised version.

Please see below a point by point response.

Reviewer 1

The article presents data on mitochondrial function under in-vitro exposure to Pravastatin and Gemfibrozil. The study is based on standard methods of respirometry and technically well conducted. In my opinion, several limitations need to be addressed more deeply:

Respirometry has been conducted at 30°C, which obviously does not correspond to physiological conditions. It cannot be simply assumed that the effects of temperature are equal for all parts of the respiratory chain. Therefore, the data may not reflect the effects of the drugs in-vivo. This limitation needs to be addressed and discussed.

Thank you for pointing this out. We are aware of the fact, that the incubation conditions do not fully mimic the in vivo situation. However, the measurement of mitochondrial oxygen consumption at 30°C represents the standard for measurements using isolated mitochondria or tissue homogenates (Kaufmann et al, Cell Mol Life Sci, 2006, Zhou and Wallace, Toxicological Science, 1999, Nadanaciva et al. Toxicology and Applied Pharmacology 2007). We did not find any information if the single complexes of the respiratory chain are differently affected by the temperature. As far as we know, there is no possibility of measuring mitochondrial respiration in vivo. Nevertheless the in vitro measurement is not performed at the physiological body temperature and it can probably not reflect the full effects of the drugs in vivo. We addressed this limitation in the discussion (page 7, line 206-208)

In contrast to the liver, which is mainly composed by hepatocytes, tissue samples from the colon are probably less homogeneous regarding the cell type. Are data available on the composition of colon tissue homogenates? This should be discussed.

This is a very important point. To our best knowledge, we are the first performing measurements in colon homogenates so there is no data about the exact composition of this tissue homogenate. As you mentioned, there are different cell types in the colon e (e.g epithelial cells, smooth muscle cells, adipocytes and others). Thus, the observed results refer to the whole organ level and cannot be assigned to a specific cell type. We added this limitation to the discussion (page 7, lines 208-210) and we consider it for our future experiments planning them with isolated enterocytes.

It is not clear, why liver and colon were investigated, but not tissues previously studied by other authors, This could allow to better compare the data. In that sense, the introduction should be improved.

We decided to investigate liver and colon as the organs mostly affected by side effects of statins and fibrates (liver enzymes elevation, gastrointestinal discomfort). It is conceivable, that these adverse events are caused by mitochondrial dysfunction. The available data about effects of statins and fibrates on mitochondrial function are inconsistent (liver) or completely lacking (colon). We decided to examine colon tissue because there are so far no data available and we consider drug effects on the colon as a clinically relevant and probably underestimated problem. We included this explanation into the introduction (page 2, lines: 50-51)

Similarly, it is not clear why Pravastatin and gemfibrozil only were tested.

We chose pravastatin and gemfibrozil as representative agents of the groups of statins and fibrates, respectively. We decided to examine these two drugs because they are on one hand considered as least mitotoxic within their groups (Nadanaciva et al. Toxicology and Applied Pharmacology 2007, Brunmair, JPET, 2004) but on the other hand still cause side effects. We added this aspect into the introduction (page 2, lines: 70-72)

It should be avoided to speak of improvement or impairment of mitochondrial function. First of all, mitochondria have several functions within the cell. In this study, however, only mitochondrial respiration has been studied. In general, an increase in mitochondrial respiration is considered as an improvement and vice versa. However, mitochondria may also dynamically respond to specific condition, and changes in oxphos capacity or RCR may simply reflect a response rather than an improvement or impairment.

Thank you for this valuable comment. We agree that the used term „mitochondrial function“ is very general and includes many aspects. Therefore, we changed the term “mitochondrial function” into “mitochondrial respiration” throughout the manuscript. Furthermore we agree that the terms „improved“ or „impaired“ could be misleading. According to your suggestion we changed the title of the manuscript, changed the terms „improved“ and „deteriorated“ mitochondrial respiration into descriptive terms like “enhanced” or “reduced” throughout the manuscript and adjusted the discussion and conclusions accordingly (page 8, lines: 238-241 and page 9, lines 313-314)

Reviewer 2 Report

One of the main concerns is the primary novelty of this work. The authors must need to find the way to improve the novelty in order to expand impact in this research field.

It is well established that statins can have serious adverse effects, which may be related to development of mitochondrial dysfunctions. Mechanistically it is still unclear how statins cause mitochondrial dysfunction. Authors should discuss the limitation of the work.

The ability of the mitochondria to make ATP and to consume oxygen in response to energy demands serves as a reliable hallmark of its functional state. Authors should measure ATP as an another way to measure mitochondrial dysfunction besides only measure oxygen consumption.

The methods section is too much confused. Please clarify. Describe in topic each measure may facilitate the understanding. Did you measure 1) Mitochondrial oxygen consumption, 2) Mitochondrial respiration, 3) Maximal mitochondrial respiration and 4) Mitochondrial leakage? If you did, please describe separately.

If the measurement was made only in tissue homogenate, authors should consider measure basal cell respiratory rate, ATP production, proton leak, maximum respiration, and spare respiratory capacity in intact cells. (Seahorse method??)

Rotenon was used to inhibit complex I and oligomycin was used to block ATP production, however authors did not present the results before and after inhibition or if treatment with statin or fibrate changed the result.

Author Response

Dear Editor, dear Reviewers,

We sincerely thank the Reviewers for evaluating our submission and their constructive criticism. We addressed all comments and have revised our paper accordingly. After implementing the Reviewer’s points we think the manuscript has greatly improved.

Additionally, we revised the manuscript carefully concerning the repetition rate with our previous publications as suggested by the Assistant Editor.

Changes made in the manuscript are highlighted in yellow in the revised version.

Please see below a point by point response.

Reviewer 2:

One of the main concerns is the primary novelty of this work. The authors must need to find the way to improve the novelty in order to expand impact in this research field.

To the best of our knowledge, we are the first examining the effects of pravastatin and gemfibrozil on colonic mitochondrial function.

Studies on intestinal mitochondria are necessary not only because of the gut being the target organ for side effects, but also as an important organ to maintain barrier function and prevent translocation of bacteria and toxins into the blood and local lymph system, thus preventing sepsis. Therefore, adequate oxygen utilisation and, as a consequence, sufficient oxygenation of the gastrointestinal mucosa are considered crucial for the prevention and therapy of critical illness. While many studies in other organs demonstrate the impact of mitochondrial function in different diseases studies on intestinal mitochondria are very rare. This might be related to the sophisticated measurement in contrast to heart or liver. We could successfully establish this method in our laboratory recently. In a previous study, we have shown for the first time the involvement of hepatic and colonic mitochondrial function in long-term model of abdominal sepsis (Herminghaus et al, ICMx, 2019) as well as effects of different drugs on colonic and hepatic mitochondrial respiration (Herminghaus et al. Front Med. 2018, Herminghaus et al. Eur J Pharmacol, 2019). This data suggest the important role of colonic mitochondria in critical illness. More understanding is necessary to detect the exact role. Nevertheless, this might be a promising therapeutic target.

Taken together, the diverging drug effects on mitochondrial function in different organs and the overall importance of intestinal mitochondrial function supports the novelty of our findings.

In our current study we investigated the effects of pravastatin and gemfibrozil (as representative agents of the statin and fibrate group respectively) on hepatic and colonic mitochondrial respiration focusing on the role of mitochondrial respiration in the pathogenesis of drug side effects. Both drugs are considered as least mitotoxic in their groups and many authors failed to show any negative effects of pravastatin on mitochondrial function. Nevertheless, these drugs can still cause gastrointestinal dysfunction and elevation of liver enzymes. We could show that pravastatin and gemfibrozil reduce mitochondrial respiration in hepatic mitochondria and gemfibrozil diminishes additionally the efficacy of OXPHOS. These results are innovative and clinically important since they potentially explain the pathogenesis of the side effects of pravastatin and gemfibrozil.

It is well established that statins can have serious adverse effects, which may be related to development of mitochondrial dysfunctions. Mechanistically it is still unclear how statins cause mitochondrial dysfunction. Authors should discuss the limitation of the work.

Thank you for this important remark. Adverse events are not fully understood and complex processes including many factors like preexisting organ damage and co-medication. In our experiments, we examined the mitochondrial oxygen consumption, which depicts only one aspect of the complex mitochondrial function within a cell. The next step is to analyze the cause of mitochondrial dysfunction. We added this aspect to the conclusions (page 9, lines 313-314)

The ability of the mitochondria to make ATP and to consume oxygen in response to energy demands serves as a reliable hallmark of its functional state. Authors should measure ATP as an another way to measure mitochondrial dysfunction besides only measure oxygen consumption.

We thank the Reviewer for this suggestion. It should be taken into account that ATP content does not accurately reflect mitochondrial status (Brand & Nicholls 2011 REVIEW, PMID: 21726199; Nicholls & Ferguson, Bioenergetics, ISBN: 978-0-12-388425-1), as ATP levels are defined not only by ATP production but also by the activity of ATP-dependent enzymes. Net cellular ATP contents are always in a steady state between synthesis and consumption. We determinated the ATP- production indirectly calculating the ADP-ratio. However, measurement of the ATP concentration simultaneously to the assessment of the oxygen consumption and the efficacy of oxidative phosphorylation could provide additional information. Unfortunately, we are unable to perform additional experiments to measure ATP-content as suggested. Measuring ATP-concentration in the tissues after treating them with different drug concentrations would demand enrolling another 48 animals, which would offend against commonly accepted '3Rs': replacement, reduction and refinement. We will consider your suggestion for our future studies.

The methods section is too much confused. Please clarify. Describe in topic each measure may facilitate the understanding. Did you measure 1) Mitochondrial oxygen consumption, 2) Mitochondrial respiration, 3) Maximal mitochondrial respiration and 4) Mitochondrial leakage? If you did, please describe separately.

Thank you for this remark. We use the terms state 2, 3 and 4 which are not consistently defined and used in the literature. We use the term state 2 based on the definition of Nichols and Fergusson ,1992:

State 2: substrate added, respiration low due to lack of ADP. .. the controlled respiration prior to addition of ADP, which is strictly termed “state 2”, is functionally the same as state 4, and the latter term is usually used for both states’ (Nicholls & Ferguson 1992).

See also: http://www.bioblast.at/index.php/LEAK_respiration#Protocols_for_measurement_of_LEAK_respiration

We used the term state 3 as ADP induced mitochondrial respiration.

See also:

http://www.bioblast.at/index.php/State_3

The exact definition of the used terms is explained in the manuscript (page 3, lines 106-118).

We clarified the method section and divided it into separate sections as suggested (page 2-3, lines: 78-147)

If the measurement was made only in tissue homogenate, authors should consider measure basal cell respiratory rate, ATP production, proton leak, maximum respiration, and spare respiratory capacity in intact cells. (Seahorse method??)

Seehorse method and measurement of oxygen consumption using Clark-electrode are both suitable methods for assessing of mitochondrial function and have both their assets and shortcomings (Divakaruni et al. Current Protocols in Toxicology, 2014). The measurement in tissue homogenates allows application of membrane impermeable substances like ADP and many mitochondrial substrates (Palmeira and Moreno, Mitochondrial Bioenergetics, 2012, ISBN: 978-1-61779-381-3). This method is well established in our laboratory (Herminghaus et al. Front Med. 2018, Herminghaus et al. Eur J Pharmacol, 2019, Herminghaus et al, ICMx, 2019).

We meausured some of the suggested parameters (proton leak=state 2, maximum respiration= state 3, ATP-production=ADP/O) (see also answer 4). Measurement in intact cells would require the separation of the cells from the tissue. This procedure is complicated but we consider it for the future experiment also to enable to measure mitochondrial respiration in separated cell populations (see also answer to point 2 of Reviewer 1).

We will consider performing our future experiments with a seahorse method, but at the moment we do not have this device to our disposal.

Rotenon was used to inhibit complex I and oligomycin was used to block ATP production, however authors did not present the results before and after inhibition or if treatment with statin or fibrate changed the result.

We appreciate this suggestion. Rotenone was added to the tissue homogenate before succinate and/or drugs were added. We highlighted in “Materials and Methods” on page 3, lines 109-110. Therefore, the presented results with different drug concentrations represent the effects of the drugs on the rotenon inhibited respiratory chain.

Oligomycin was added as a quality control to check if the inner mitochondrial membrane was intact after the preparation procedure. We continued our experiments only if O2 consumption after adding oligomycin was similar to state 2. Therefore we can consider state 2 as an indicator for proton leak. The quality control took place at the beginning of every experiment. We clarified it in “Materials and Methods “on page 3, line 127. State 2 was assessed in the presence and absence of drugs. We can provide the results for state 2 as supplement if required.

Round 2

Reviewer 1 Report

The current version of the manuscript includes the description and discussion of some limitations according to my previous comments. I have no further concern with the manuscript

Reviewer 2 Report

The authors' response to the comments were satisfactory, well thought and layed out. The methods section were also improved.